# Regulatory B Cells—Immunopathological and Prognostic Potential in Humans

**DOI:** 10.3390/cells13040357

**Published:** 2024-02-18

**Authors:** Johanna Veh, Carolin Ludwig, Hubert Schrezenmeier, Bernd Jahrsdörfer

**Affiliations:** 1Institute for Transfusion Medicine, Ulm University Hospitals and Clinics, 89081 Ulm, Germany; 2Institute for Clinical Transfusion Medicine and Immunogenetics, German Red Cross Blood Donation Service Baden-Württemberg-Hessen, 89081 Ulm, Germany

**Keywords:** regulatory B cell, Interleukin 10, GraB cell, granzyme B, immunosuppression

## Abstract

The aim of the following review is to shed light on the putative role of regulatory B cells (Bregs) in various *human* diseases and highlight their potential prognostic and therapeutic relevance in humans. Regulatory B cells are a heterogeneous group of B lymphocytes capable of suppressing inflammatory immune reactions. In this way, Bregs contribute to the maintenance of tolerance and immune homeostasis by limiting ongoing immune reactions temporally and spatially. Bregs play an important role in attenuating pathological inflammatory reactions that can be associated with transplant rejection, graft-versus-host disease, autoimmune diseases and allergies but also with infectious, neoplastic and metabolic diseases. Early studies of Bregs identified IL-10 as an important functional molecule, so the IL-10-secreting murine B10 cell is still considered a prototype Breg, and IL-10 has long been central to the search for human Breg equivalents. However, over the past two decades, other molecules that may contribute to the immunosuppressive function of Bregs have been discovered, some of which are only present in human Bregs. This expanded arsenal includes several anti-inflammatory cytokines, such as IL-35 and TGF-β, but also enzymes such as CD39/CD73, granzyme B and IDO as well as cell surface proteins including PD-L1, CD1d and CD25. In summary, the present review illustrates in a concise and comprehensive manner that although human Bregs share common functional immunosuppressive features leading to a prominent role in various human immunpathologies, they are composed of a pool of different B cell types with rather heterogeneous phenotypic and transcriptional properties.

## 1. Introduction 

The first reports on the immunoregulatory activity of B cells date back to the 1970s. At that time, B cell-depleted splenocytes isolated from ovalbumin-sensitized guinea pigs were shown to have a significantly reduced suppressive capacity on delayed-type hypersensitivity reactions in recipient animals compared to non-B cell-depleted splenocytes [1]. These early studies were important in demonstrating that B cells can suppress inflammation through the production of IL-10 but provided limited insight into the origin and phenotype of such B cells. Moreover, as these studies were conducted exclusively in animal models, it initially remained unclear to what extent the findings could be transferred to humans. The first indications that B cells can also exert immunoregulatory functions in humans came from several case reports of patients treated with the B cell-depleting antibody Rituximab. In these patients, B cell depletion was associated with the development of psoriasis [2,3] or a worsening of the course of ulcerative colitis [4]. 

Over the last three decades, the role of regulatory B cells (Bregs) has been documented in more detail in the context of different diseases [5,6]. In various animal models, Bregs have been shown to suppress autoimmune responses such as in experimental autoimmune encephalomyelitis [7], collagen-induced arthritis [8] and spontaneous colitis [9]. Bregs have also been shown to play a role in allergies [10], transplant rejection [11], infections [12], neoplastic diseases [13] and even in the course of inflammatory responses in chronic metabolic diseases [14]. Although the initial studies in the early 2000s initially attributed this immunomodulation exclusively to IL-10 [9,15,16], alternative suppression modes have more recently been discovered, especially in human pathologies, which make IL-10 a prominent, but by far not the only, functional molecule of immunosuppression in Bregs [6,17]. Since many reviews in the field do not clearly differentiate between findings made in humans and findings made in animal models, we felt the necessity to prepare a review, which focuses on the putative role of regulatory B cells (Bregs) in various *human* diseases and highlights their potential prognostic and therapeutic relevance in humans. 

## 2. Diversity of the Breg Universe

For a long time, a regulatory B cell was synonymous with an IL-10-secreting B cell. Although less is known about the development of IL-10-producing Bregs (B10 cells) in humans than in mice, in 2011, Iwata and colleagues identified CD19^+^CD24^hi^CD27^+^ B cells as the human equivalent of B10 cells in mice. They were able to show that in vitro stimulation of this B cell subpopulation with recombinant CD40 ligand and toll-like receptor agonists promoted maturation into IL-10-producing B10 cells [18]. Numerous studies in mice and humans have since identified further subgroups of B cells with immunoregulatory function. In addition to obvious species-specific differences between mice and humans, the subgroups also differ considerably in their phenotypic characteristics and the suppressive molecules that they express, although there is some overlap. The most important subtypes of Bregs currently known in humans are illustrated in Figure 1 and summarized in Table 1. Currently, four secreted molecules are thought to be central to the contact-independent immunosuppressive effects of Bregs—the three cytokines IL-10, IL-35 and TGF-β as well as the enzyme granzyme B (GrB) (Figure 1). As if this were not enough possibilities, three different isoforms exist for TGF-β alone, which bind to different receptors and can thus activate different signaling pathways. Depending on other factors, such as tissue type and environment, these different signaling pathways can have a major impact on which cellular functions are ultimately influenced by TGF-β-secreting Bregs and how [19]. In addition, the interaction between the cytokines secreted by Bregs and their putative target cells is also influenced by the ability of B cells to produce antigen-specific immunoglobulins of different classes and subclasses, such as the “proallergenic” IgE or the more “tolerogenic” IgG4. Here, too, differences can have a major influence on the development and progression of certain allergic and autoimmune diseases in humans [20].

Even the above selection leaves out a whole range of effective mechanisms that require direct cell-to-cell contact between Bregs and target cells, so the currently known immunosuppressive arsenal of Bregs has already reached an impressive diversity over the last two decades (Figure 2 and Table 1). In this context, it should also be noted that there is currently no clear consensus on the classification and definition of Bregs. Overall, the phenotypic diversity of Bregs suggests that, in principle, any B cell, regardless of its developmental stage or location, is capable of establishing an immunoregulatory potential, possibly induced by one or more yet-to-be-identified master transcription factors and by specific environmental conditions [6,17,21,22]. The fact that Bregs are found among B cell populations at different stages of maturation and differentiation suggests that a wide variety of B cell lineages can be induced to adopt a regulatory function in response to different environmental factors. However, also due to the heterogeneity of Breg subgroups, the identification of a Breg-specific transcription factor has so far been unsuccessful [6,17,21,22]. It, therefore, appears likely that Bregs do not actually represent an independent differentiation lineage of B cells but that various regulatory functional phenotypes can be induced in different B cell subsets. The molecules involved in the induction of Bregs include cytokines such as IL-21 or IL-10 [21,23,24], the B cell receptor (BCR) or downstream activators such as PMA [25] and ligands binding to various pattern recognition receptors, including Toll-like receptor ligands [21,26,27].

**Figure 1 cells-13-00357-f001:**
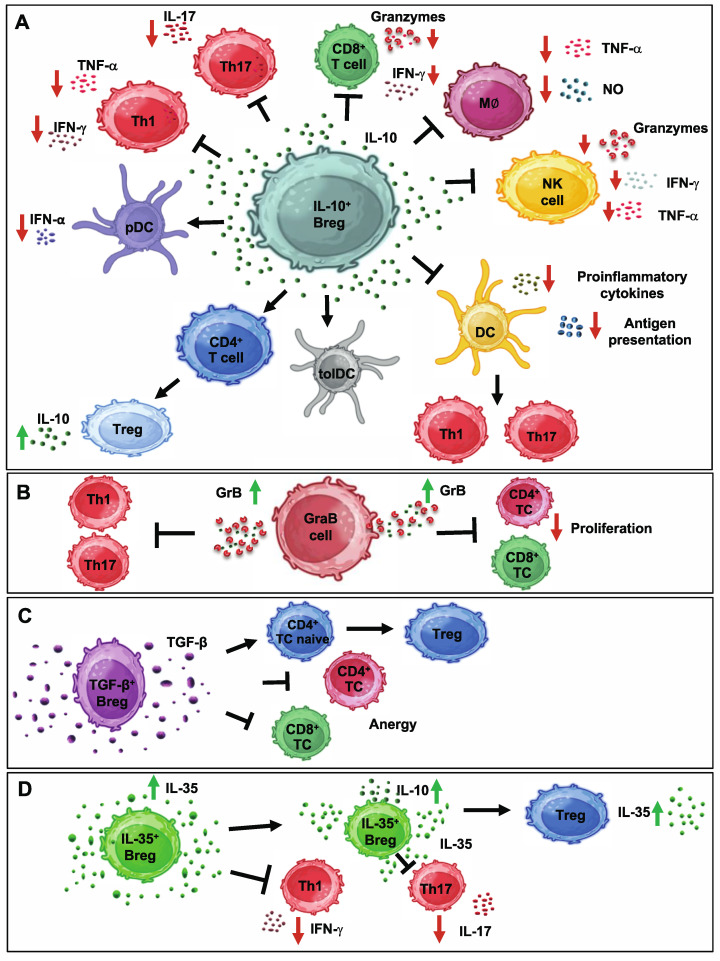
Main subtypes of human regulatory B cells. (**A**) IL-10+ regulatory B cells (Bregs) inhibit the Th1, Th17 and CD8+ T cell response, convert naïve CD4+ T cells into regulatory T cell populations and modulate pro-inflammatory cells of the innate immune system, such as macrophages, NK cells and dendritic cells via secretion of IL-10. (**B**) GrB+ regulatory B cells (GraB cells) inhibit the response of Th1 and Th17 cells and reduce the proliferation of T cells by reducing their proliferation through GrB-mediated enzymatic cleavage of the ζ-chain of their T cell receptor. (**C**) TGF-β+ Bregs act similarly to IL-10+ Bregs on naïve CD4+ T cells, generating FoxP3+ Tregs and inducing anergy in CD4+ and CD8+ T cells. (**D**) IL-35+ Bregs can promote tolerance in the context of chronic infections by supporting both IL-35-producing Tregs and their own generation. Abbreviations: Breg = regulatory B cell, CD = cluster of differentiation, DC = dendritic cell, GraB cell = GrB-secreting regulatory B cell, GrB = granzyme B, IFN = interferon, IL = interleukin, M∅ = macrophage, NK cell = natural killer cell, NO = nitric oxide, pDC = plasmacytoid dendritic cell, TGF = transforming growth factor, Th = T helper cell, TNF = tumor necrosis factor, tolDC = tolerogenic dendritic cell, Treg = regulatory T cell, ↑ = upregulation, ↓ = downregulation. Figures were prepared using BioRender (Agreement number: UA2659WLGL).

**Figure 2 cells-13-00357-f002:**
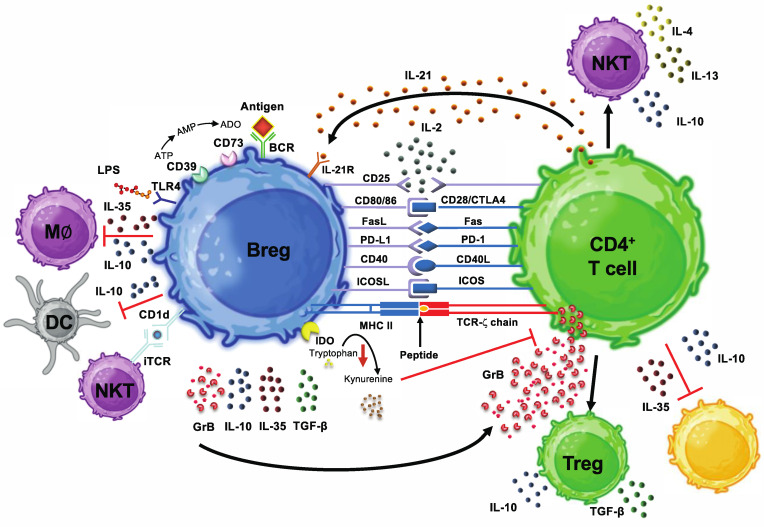
Suppressive mechanisms of human regulatory B cells. Like regulatory T cells, the currently known subpopulations of human regulatory B cells (Bregs) utilize numerous different mechanisms for their immunosuppressive activity. These include the secretion of soluble molecules such as the cytokines IL-10, IL-35 or TGF-β as well as the cytotoxic enzyme granzyme B (GrB). On the other hand, molecules expressed on the surface of Bregs also play a major role in inhibiting immune responses. These molecules can be enzymes such as CD39, CD73 or IDO, which convert certain substrates in such a way that this results in the inhibition of certain immune cells. On the other hand, molecules that require direct contact with corresponding complementary ligands on other immune cells are also involved in the immunosuppressive effect of Bregs. These include antigen-presenting molecules such as CD1d or MHC II, costimulatory molecules such as CD80, CD86 or CD40 or ligands for so-called “death receptors” such as Fas or PD-1. Key inducers of Bregs include cytokines such as IL-21 and IL-10, the B cell receptor (BCR) and Toll-like receptors (TLRs). Abbreviations: ADO = adenosine, AMP = adenosine monophosphate, ATP = adenosine triphosphate, BCR = B cell receptor, Breg = regulatory B cell, CD = cluster of differentiation, DC = dendritic cell, FasL = Fas ligand, GrB = granzyme B, ICOSL = inducible costimulator ligand, IDO = indoleamine 2,3-dioxygenase, IFN = interferon, IL = interleukin, iTCR = invariant T cell receptor, M∅ = macrophage, MHC = major histocompatibility complex, NK cell = natural killer cell, NKT = natural killer T cell, PD-L1 = programmed cell death ligand-1, TCR = T cell receptor, TGF = transforming growth factor, Th = T helper cell, TLR = toll-like receptor, TNF = tumor necrosis factor, Treg = regulatory T cell, ↓ = downregulation. Figures were generated using BioRender (Agreement number: UA2659WLGL).

**Table 1 cells-13-00357-t001:** Currently known subtypes of human regulatory B cells and their key immunosuppressive mechanisms.

Breg Type	Phenotype	Locations	Immunosuppressive Molecules	References
Granzyme B^+^ B cells (GraB cells) *	CD19^+^ CD20^+^ GrB^+^ CD86^+^ CD147^+^, IDO^+^, (CD38^±^ CD25^±^ CD27^+^ CD1d^±^ CD5^±^ CD10^+^ IgM^±^) **	Peripheral blood, Solid tumors	Granzyme B >> TCR-ζ cleavage Granzyme B, IDO, CD25	[23,26,28,29] [24]
B10 cells	CD19^+^ CD24^hi^ CD27^+^	Spleen, peripheral blood, gastric mucosa, gastric carcinoma	IL-10	[18,30,31,32,33]
Immature transitional B cells *	CD19^+^ CD24^hi^ CD38^hi^ CD1d^hi^	Peripheral blood, liver	IL-10, CD80/86	[34,35,36,37,38,39]
CD5^+^ B cells	CD19^+^ CD5^+^ GrB^+^ CD1d^hi^	Peripheral blood	Granzyme B IL-10	[40] [41]
Tim-1^+^ B cells	CD19^+^ Tim-1^+^	Spleen, peripheral blood	IL-10	[42,43,44]
Adipose tissue B cells	CD19^+^ CD27^+^ CD38^hi^	Adipose tissue	IL-10	[45,46]
Br1 cells *	CD19^+^ CD25^+^ CD71^hi^ CD73^lo^	Peripheral blood	IL-10, IgG4	[47]
CD73^+^ B cells	CD19^+^ CD39^+^ CD73^+^	Spleen, peripheral blood	CD39/CD73 >> AMP/Adenosin	[48,49]
Plasmablasts	CD19^+^ CD24^hi^ CD27^int^ CD38^+^ CD138^+^ IgA^+^ PD-L1^−^ IL-10^+^	Lymph nodes, peripheral blood, spleen	IL-10 TGF-β	[50] [51]
PD-L1^hi^ B cells	CD19^+^ PD-L1^hi^	Spleen, solid tumors	PD-L1, IgA, IL-10	[52,53,54]
CD9^+^ B cells	CD19^+^ CD9^+^	Spleen, peripheral blood	IL-10	[55]
* No cellular equivalent in the mouse	** (±) Inconsistent expression on GraB cells			

Three Breg subtypes listed here have not yet been detected in animal models. On the other hand, a number of Breg subtypes have been described in the mouse have not (yet) been detected in humans. * No cellular equivalent in the mouse, ** (±) Inconsistent expression on GraB cells. Abbreviations: B10 cell = IL-10-secreting Breg, Br1 = Breg type 1, Breg = regulatory B cell, CD = cluster of differentiation, GrB = granzyme B, IDO = indoleamine 2,3-dioxygenase, Ig = immunoglobulin, IL = interleukin, PD-L1 = programmed cell death ligand 1, TCR = T cell receptor, TGF = transforming growth factor.

## 3. Regulatory B Cells in Inflammatory Immunopathologies

A healthy immune system is characterized by a fine balance between pro-inflammatory responses on the one hand, which should enable the efficient elimination of infectious agents and malignant cells, and anti-inflammatory responses on the other hand, which should prevent the development of chronic inflammation and autoimmunity. B cells contribute to a pro-inflammatory immune response through antibody production, antigen presentation and the production of certain cytokines. In the field of clinical transplant medicine, for example, the presence of B cells has traditionally been associated with an unfavorable prognosis due to their ability to differentiate into plasma cells and present antigens to T cells, potentially playing a role in both acute antibody-mediated and chronic transplant rejection [56]. In light of these findings, Menna Clatworthy and colleagues reported in 2009 an interesting observation when comparing two different induction regimens (B cell depletion with anti-CD20 vs. the conventional regimen with anti-CD25 antibodies) in non-sensitized patients undergoing renal transplantation [57]. The authors observed a higher incidence of acute rejection in B cell-depleted patients (5/6) compared to those who received conventional anti-CD25 induction (0/7). The study was prematurely terminated after enrollment of only 13 patients, and the authors speculated that the results may be due to anti-CD20-mediated depletion of Bregs previously described only in mouse models. 

Only a few months later, Claudia Mauri’s group described for the first time the existence of regulatory B cells in humans and their functional deficiency in patients with systemic lupus erythematosus [34]. In the same year, three independent studies demonstrated a further link between B cells and transplant tolerance in kidney transplant recipients. The first study was led by the group of Sophie Brouard and showed that tolerant recipients exhibited an increased number of peripheral B cells, which were characterized by a defect of the IL-21-dependent differentiation into plasma cells [58,59] and which instead were stoodk out by the expression of the cytotoxic enzyme GrB [28,60,61,62]. Two further multicenter studies from the US and Europe [63,64], published another few months later, were able to demonstrate an increased frequency of B cells in tolerant kidney transplant recipients. Despite some differences between the three studies, the overall conclusion was that an increased number of B cells is a reproducible parameter in tolerant kidney transplant patients (Table 2). 

Although there is strong overall evidence that Bregs are centrally involved in the induction and maintenance of tolerance in various immunopathologies, there are currently no therapeutic approaches that involve an intentional modulation of the frequency or activity of Bregs. Most studies on chronic inflammatory responses such as transplant rejection, autoimmune diseases, chronic inflammatory diseases and allergies have demonstrated a lack or a functional deficit of circulating Bregs (Table 2, Table 3 and Table 4).

Reduced frequencies or insufficient suppressive capacity of Bregs have been reported in patients with various autoimmune diseases, chronic GvHD or asthma as well as in patients with food allergies. Conversely, the limited data from studies on allergen immunotherapy (AIT) indicate that the number of Bregs detectable in the blood increases during AIT. These results therefore are in line with the above-described findings from transplant medicine that certain patients after kidney transplantation exhibit a GrB^+^ Breg phenotype, which appears to confer long-term tolerance to the transplant and which is lacking in patients with transplant rejection. Nevertheless, it must be noted that no causality between the individual parameters has yet been established, and it, therefore, remains unclear for the time being whether tolerance is actually induced by Bregs or whether the frequency of these cells increases as a result of tolerance. Ultimately, only clinical studies with an adoptive transfer of ex-vivo-generated human Bregs will be able to provide an answer to this question.

## 4. Regulatory B Cells in Infections

One of the most important tasks of B cells during infections is the production of antigen-specific antibodies and the formation of memory B cells. However, in addition to the specific response via the B cell receptor, microbial products such as nucleic acids and cell wall components can also activate a broader panel of immune cells via Toll-like receptors (TLRs). This activation can involve B cells and Bregs [121,122,123] but also cells of the innate immune system including monocytes or macrophages, which are then stimulated to phagocytose and express pro-inflammatory cytokines such as IL-1β and TNF-α [124,125,126]. In particular, chronic systemic infections can trigger sustained immunosuppression [127]. In these cases, the induction of Bregs can be regarded as an example of microbial escape mechanisms prior to the mounting of a protective immune response of the host. Therefore, chronic infections with bacteria, viruses, fungi and parasites are typically associated with increased Breg frequencies, which is in contrast to most acute pro-inflammatory conditions (Table 5). As an example, this can be observed in infections with Mycobacterium tuberculosis, where IL-10- and IL-35-producing B cells are induced during the active stage of pulmonary tuberculosis [128].

Also in viral infections, various observations indicate that active immune responses to viral proteins and nucleic acids are mounted in the blood, spleen and lymph nodes, which involve the induction of Bregs. During chronic infection with the hepatitis B virus (HBV), for example, a higher number of IL-10-producing CD19^+^CD24^hi^CD38^hi^ regulatory B cells are found in patients than in healthy controls [36]. In patients with acute infections by the human immunodeficiency virus (HIV), we showed that the HIV protein Nef induces T helper cells to secrete the cytokine IL-21 without expressing relevant amounts of the CD40 ligand [29]. As a result, such incompletely activated T helper cells induce the differentiation of a large percentage of B cells into GrB-secreting GraB cells, both in vitro and in vivo, when they come into contact with each other [29]. If the missing CD40 stimulus is added in vitro to cell cultures in the form of soluble CD40L multimers, the corresponding B cells differentiate into antibody-producing plasma cells instead [29]. Similar mechanisms can be found in patients with congenital errors of immunity, which often are associated with numerical and functional aberrations of B cells, including regulatory B cells [139]. We described an immunodeficient patient who lacked intact CD40 signaling due to a congenital NEMO mutation [29]. Accordingly, a large proportion of peripheral B cells (up to 100%) in this patient were GrB^+^ GraB cells [29]. It is important to note at this point that GraB cells are only observed in humans and not in mice [140]. GraB cells therefore occupy a special position among the Bregs previously described in humans with regard to their development and phenotype (Figure 3). This is also underlined by the fact that, as described above, GraB cells appear to play an important role in kidney-transplanted tolerant patients, where the differentiation of B cells towards plasma cells is obviously shifted in favor of development into GrB-secreting Bregs [28,60,61,62].

## 5. Regulatory B Cells in Neoplastic Diseases

Tertiary lymphoid structures (TLS), which contain aggregates of B and T lymphocytes within the tumor stroma, have recently attracted considerable attention as a prognostic and predictive factor [141,142,143]. The presence of activated B cells in TLS has been associated with an improved clinical outcome and enhanced response to immunotherapy in various tumor types [143,144,145,146,147,148,149,150]. Therefore, the overall evidence suggests that B cells play a rather favorable prognostic role for most tumor entities [151] so the main goal of potential immunotherapeutic interventions may rather be an enhancement of their activity instead of a suppression.

Nevertheless, in some tumor entities, an immunosuppressive role of B cells has been observed as well [151]. For example, increased expression of B cell and plasma cell gene signatures has been associated with poor clinical prognosis in glioblastoma [152,153,154]. Similarly, increased expression of B cell-typical genes is also correlated with poor survival in clear cell renal carcinoma [152,155]. In ductal breast carcinoma, immunohistochemical analysis revealed that a high proportion of CD138^+^ plasma cells was associated with a low survival rate [156]. In addition, in various gynecological tumor entities such as breast, cervical and ovarian cancer, Bregs of the GrB-secreting type (GraB cells) were found in close proximity to IL-21^+^ T cells [24]. On the other hand, B cell gene expression signatures have also been correlated with improved metastasis-free survival in gynecological tumors [157]. Moreover, analyses of tissue samples from HER2^+^ and triple-negative patients with breast cancer support the association between an increased frequency of tumor-infiltrating B cells and an improved clinical outcome [158]. Therefore, the overall clinical data suggest that B cells capable of antigen presentation, cytokine production and antibody-mediated cellular cytotoxicity may contribute to an antitumor response, while the presence of immunosuppressive B cell phenotypes and antibody isotypes such as IgG4 are prognostically rather associated with tumor-promoting effects [159] (Table 6).

## 6. Conclusions

After several decades of research, it has become clear that Bregs possess a significant prognostic value for a large variety of diseases. While Bregs may play a protective role in diseases with undesired and excessive immune responses, such as autoimmune diseases, transplant rejection, graft-versus-host disease or allergies, their induction or overactivation during infections or tumor diseases may rather negatively affect disease courses. Currently, there are no clinical approaches available that result in an intentional modulation of the Breg number or functionality, neither by in vivo inhibition, depletion or induction nor by adoptive transfer of ex vivo-manufactured Bregs. The conduct of clinical studies in humans is complicated by the fact that due to species differences, at least for certain Breg subpopulations, the predictive value of animal models is rather limited [140]. Notwithstanding these hurdles, Bregs may potentially be a very exciting and promising addition to the growing arsenal of novel immune cell therapeutics in the form of *Advanced Therapy Medicinal Products*. A current online search (accessed on 31 January 2024, https://clinicaltrials.gov/) for completed or currently ongoing clinical trials revealed only descriptive studies on the presence and potential role of Bregs in certain diseases but not a single interventional trial using cellular therapeutics based on Bregs. While this clearly shows the existing research gap in the field, the time appears ripe for the implementation of GMP-compliant manufacturing processes for certain types of autologous or HLA-matched allogeneic Bregs. In the end, evidence for their therapeutic potential in humans can be provided only when patients with severe diseases start getting access to GMP-compliant Bregs, beginning with their compassionate use within hospital exemptions or small phase I clinical trials.

## Figures and Tables

**Figure 3 cells-13-00357-f003:**
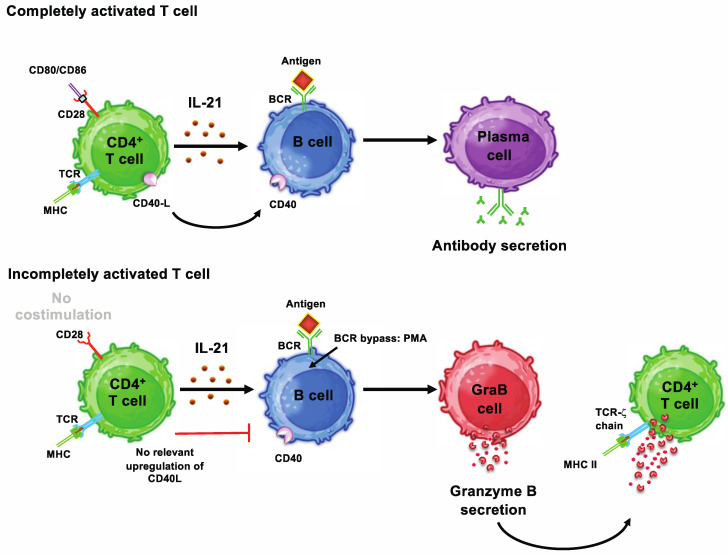
Development of granzyme B-secreting regulatory B cells as deviation of their IL-21-dependent differentiation into plasma cells. Complete activation of CD4^+^ T cells requires simultaneous stimulation of the T cell receptor (TCR) via MHC/peptide complexes and of CD28 by costimulatory molecules such as CD80 or CD86 on antigen-presenting cells (APC). When T cells are fully activated, they secrete IL-21 and express large amounts of CD40L on their surface, which enables them to initiate differentiation of B cells into antibody-producing plasma cells. In certain situations, the TCR of CD4^+^ T cells can be activated without simultaneous co-stimulation via CD28. As an example, the HIV protein Nef is able to directly activate T cells via the TCR in the absence of professional APC. This results in incomplete activation of T helper cells, which react with secretion of IL-21 but no relevant upregulation of CD40L. After interaction with such incompletely activated T helper cells, BCR-stimulated B cells develop into GrB^+^ B cells with regulatory potential (GraB cells) instead of plasma cells. Abbreviations: BCR = B cell receptor, CD = cluster of differentiation, GraB cell = GrB-secreting regulatory B cell, GrB = granzyme B, IL = interleukin, MHC = major histocompatibility complex, PMA = Phorbol-12-Myristate-13-Acetate. Figures were generated using BioRender (Agreement number: UA2659WLGL).

**Table 2 cells-13-00357-t002:** Transplant Rejection/Tolerance. Immunopathological and prognostic relevance of regulatory B cells in humans. Listed are various diseases and immunotherapeutic interventions in which inflammatory responses associated with regulatory B cells have been described in humans.

Observations with Regard to Regulatory B Cells	References
Correlation of B cell-depleting therapy with impaired graft survival	[57]
Increased frequency of Bregs with memory phenotype and expression of costimulatory molecules such as CD80, CD86, CD40 and CD62	[61]
Tolerance to allografts due to IL-10-producing Bregs	[60,63,64,65,66,67]
Correlation of low IL-10/TNF-α ratios with poor graft outcome due to lack of inhibition of Th1-type cytokine production by transitional B cells	[68]
Inhibition of CD4^+^CD25^+^ effector T cells by IL-21-dependent granzyme B-expressing Bregs and resulting tolerance after kidney transplantation	[28,69]
Association of transitional IL-10^+^ B cells with renal transplant tolerance	[70,71]
B cells activate autocrine IL-10 signaling pathway in CD4^+^ T cells in response to IFN-γ production, disruption of this mechanism causes increased IFN-γ production	[72,73]
Ratio of transitional T1/T2 cells as a predictor of kidney graft function. Increased IL-10 secretion and decreased TNF-α secretion by Th1 cells	[74]
Identification of tolerant patients based on genetic characteristics (AKR1C3, CD40, CTLA4, ID3, MZB1, TCL1A) and clinical parameters	[75]

**Table 3 cells-13-00357-t003:** Immunopathological and prognostic relevance of regulatory B cells in humans autoimmune diseases and chronic inflammatory diseases. Listed are various diseases and immunotherapeutic interventions in which inflammatory responses associated with regulatory B cells have been described in humans.

	Observations with Regard to Regulatory B Cells	References
Multiple Sclerosis (MS)	Reduced IL-10 production by Bregs	[76]
	Quantitative deficit of Bregs	[77,78]
	Association of treatment success with numerical increase of Bregs in MS	[77,78,79,80,81]
Rheumatoid Arthritis (RA)	Lack of IL-10-producing B cells	[82,83]
	Lack of PD-L1-expressing B cells	[84]
	Conversion of naive T cells into Th1 cells by B10 cells	[85]
	Suppression of RA activity via aryl-hydrocarbon receptor on Bregs	[86]
Systemic lupus erythematosus (SLE)	Functional impairment of the inhibitory effect of Bregs in SLE patients	[34,87]
	Suppression of the expansion of granzyme B-secreting CD5^+^ B cells by IL-21 in SLE patients	[40]
	Lack of IL-10^+^ and IL-35^+^ B cells in SLE patients	[88]
Sjögren’s syndrome	Restricted IL-10 production and TfH cell suppression	[89]
Systemic sclerosis	Induction of IL-10^+^ Bregs after application of mesenchymal stromal cells in systemic sclerosis	[90]
Periodontitis	Review of therapeutic application approaches of Bregs and Tregs in periodontitis	[91]
Chronic inflammatory bowel diseases	Reduced IL-35 expression on regulatory T and B cells in patients with inflammatory bowel disease	[92,93]
Graft-versus-host disease (GvHD)	IL-10 is a significant prognostic factor with regard to the suppression of GvHD	[94]
	Bregs from patients with cGVHD produce less IL-10 than Bregs from healthy donors and patients without cGVHD	[95]
	Lack of Bregs (CD24^+^^+^CD271 und CD27^+^^+^CD38^+^^+^ plasmablastic B cells) and defective IL-10 production correlate with cGVHD severity	[96]
	Occurrence of chronic GvHD associated with significantly reduced frequencies of transitional CD10^+^ B cells	[97]
	The severity of cGvHD correlates with a lack of Bregs	[98]

**Table 4 cells-13-00357-t004:** Immunopathological and prognostic relevance of regulatory B cells in humans allergic diseases. Listed are various diseases and immunotherapeutic interventions in which inflammatory responses associated with regulatory B cells have been described in humans.

	Observations with Regard to Regulatory B Cells	References
Bronchial asthma	Numerical deficit of CD24^+^^+^CD27^+^ Bregs, impaired IL-10 secretion under LPS stimulation	[99,100,101]
	Induction of apoptosis in CD3^+^CD4^+^CD25^+^ effector T cells by CD9^+^ Bregs	[102]
	Number of RSV-infected Bregs correlates with increased viral load and reduced number of Th1 memory cells in the blood	[103]
	Lack of CD1d^+^CD5^+^ B cells in patients with allergic asthma	[104]
	Decreased frequency of regulatory B cells in pediatric patients with bronchial asthma	[105]
Allergic rhinitis	Increased frequency of CD19^+^CD24hiCD27^+^ Bregs, decreased frequency of CD19^+^CD24hiCD38hi, CD19^+^CD25^+^CD71^+^CD73 and CD19^+^CD5hiCD1d^+^ Bregs	[106,107,108,109]
	Lower frequency of IL-10-producing CD19^+^CD25^+^CD71^+^ Bregs after TLR9 stimulation	[100]
Food allergy	Decreased number of TGF-β^+^CD19^+^CD5^+^ and CD19^+^CD5^+^Foxp3^+^ Bregs in patients with cow’s milk allergy	[110,111,112]
	Increased frequency of CD19^+^CD5^+^ B cells in cow milk tolerant individuals compared to individuals with cow’s milk allergy	[113]
	Impairment of CD19^+^CD25^+^CD71^+^ Bregs with regard to IL-10 secretion	[114]
Atopic dermatitis	Deficit in CD24^+^^+^CD38^+^^+^-Bregs >> inverse correlation between Breg number and severity of atopic dermatitis	[115,116]
Allergen immunotherapy (AIT)	Increased frequency of IL-10^+^ IgG4^+^ bee venom allergen-specific B cells in non-allergic beekeepers and allergic patients after AIT	[47]
	Increased frequency of CD25^+^CD71^+^IL-10^+^ Bregs in beekeepers and allergic patients after AIT	[117]
	HDM-AIT therapy is associated with increased number of IL-10^+^ and/or IL-1RA^+^ Bregs	[118]
	LPP immunotherapy induces Bregs in patients with rhinoconjunctivitis with or without asthma	[119]
	Subcutaneous immunotherapy with grass pollen (SCIT) associated with induction of IgG4 in serum, number of IL-10^+^ Bregs increased during season in active group	[120]

**Table 5 cells-13-00357-t005:** Immunopathological and prognostic relevance of regulatory B cells in humans infections. Listed are various diseases and immunotherapeutic interventions in which inflammatory responses associated with regulatory B cells have been described in humans.

	Observations with Regard to Regulatory B Cells	References
**Viral Infections**		
FSME (Vaccination)	GrB secretion by Bregs as an early cellular immune response to inhibit viral replication	[23]
HBV	Increased proportion of IL-10-producing B cells in chronic hepatitis B	[36]
HIV	Association of HIV with PDL1^+^ Bregs and increased IL-10 production	[129]
	Inhibition of T cell proliferation via degradation of the TCR-ζ chain by granzyme B-expressing GraB cells	[29]
	Expression of immunosuppressive cytokines (IL-10, TGF-β, IL-35) by CD19^+^CD24^+^^+^CD38^+^^+^-Bregs after contact with HIV-1 particles	[130]
EBV	Increased granzyme B secretion by B cells after acute infection with Ebstein-Barr virus (EBV)	[131]
RSV	Number of RSV-infected Bregs correlates with increased viral load and reduced number of Th1 memory cells in the blood	[103]
**Bacterial Infections**		
Mycobacterium tuberculosis	Induction of CD19^+^CD1d^+^CD5^+^ Bregs and IL-10^+^ IL-35^+^ Bregs by Mycobacterium tuberculosis with subsequent suppression of Th17 cells	[41,128]
Helicobacter pylori	Suppression of CD24^+^CD38^+^ Bregs after infections with Helicobacter pylori	[132]
	Induction of IL-10^+^ BC by Helicobacter pylori	[133]
**Helminthoses**		
Trypanosoma cruzi, Paracoccidioides brasiliensis	Increased IL-10 production by Bregs in MS patients with helminthoses, association with less severe course of MS	[134]
Schistosoma haematobium	Increased frequency of IL-10-producing B cells in infections with Schistosoma haematobium	[135]
Wuchereria bancrofti	Increased frequency of Bregs and Tregs in infections with W. bancrofti with positive influence on their survival by IL-10	[136]
**Parasitoses**		
Leishmania	IL-10-mediated regulation of T cells by Bregs in Leishmania infantum	[137]
	T cell suppression by regulatory B10 cells in visceral leishmaniasis with overproduction of IgD, IL-10 and PDL1	[138]

**Table 6 cells-13-00357-t006:** Neoplastic Diseases. Immunopathological and prognostic relevance of regulatory B cells in humans. Listed are various diseases and immunotherapeutic interventions in which inflammatory responses associated with regulatory B cells have been described in humans.

Observations with Regard to Regulatory B Cells	References
Breg-associated secretion of IgG4 promotes tumor growth	[160,161]
Granzyme B-expressing B cells in the microenvironment of various gynecological tumor diseases support immune tolerance to tumor antigens	[24]
Infiltration of tumors by Bregs as a negative predictor in breast, gastric, esophageal and squamous cell carcinomas in the head and neck region	[31,162,163,164,165]
CD19^+^CD24^+^CD38^+^ Bregs as a negative prognostic factor in acute myeloid leukemia (AML)	[166]
Increased IL-10 production by CD19^+^CD24^+^^+^CD27^+^ B cells in gastric carcinoma	[31]
STING-induced IL-35^+^ Bregs suppress NK cells in pancreas carcinoma	[167]
Increased number of CD19^+^IL-10^+^ Bregs after surgery in patients with hepatocellular carcinoma (HCC)	[168]
Positive correlation between numbers of Bregs, regulatory T cells and TH17 cells as well as concentration of IL-10, IL-35 and BAFF in patients with primary hepatocellular carcinoma (PHC)	[169,170]

## Data Availability

Not applicable.

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
