# Peer review of "Regulatory B Cells—Immunopathological and Prognostic Potential in Humans"

_cells, 2024, doi:10.3390/cells13040357_

Round 1

Reviewer 1 Report

Comments and Suggestions for Authors

The authors in this review discuss a very important issue which is the immunopathological role and prognostic potential of regulatory B cells in different human diseases. The article is well-written, showed adequate illustrations and is very organized. Minor comment to be addressed:

Please add these references to table 2, which points to the role of regulatory B cells in HCC: doi: 10.1186/1479-5876-10-14., doi: 10.1111/cei.13094 , doi: 10.3390/vaccines8030380.

Author Response

Reviewer #1:

The authors in this review discuss a very important issue which is the immunopathological role

and prognostic potential of regulatory B cells in different human diseases. The article is wellwritten,

showed adequate illustrations and is very organized. Minor comment to be addressed:

Please add these references to table 2, which points to the role of regulatory B cells in HCC: doi:

10.1186/1479-5876-10-14., doi: 10.1111/cei.13094, doi: 10.3390/vaccines8030380.

Reference has been added.

Reviewer 2 Report

Comments and Suggestions for Authors

Interesting review on regulatory B cells, with special focus in human Bregs.

Minor requests:

Please, note that greek letters are not visible along the document. They are only visible in the Figures.

The size of the letters in Tables 1 and 2 is too small. Please, increase their size. It may help to split table 2 in two, with less lines per table, and with similar letter size than in the regular text.

Author Response

Reviewer #2:

Interesting review on regulatory B cells, with special focus in human Bregs.

Minor requests:

Please, note that greek letters are not visible along the document. They are only visible in the

Figures.

Letters have been corrected.

The size of the letters in Tables 1 and 2 is too small. Please, increase their size. It may help to

split table 2 in two, with less lines per table, and with similar letter size than in the regular text.

Letter size in tables has been increased, table 2 has been split in 5 sub-tables. If letter size is still too small tables could be turned into an upright position.

Reviewer 3 Report

Comments and Suggestions for Authors

In the present work, Veh et al. try to review regulatory B cells - immunopathological and prognostic potential in humans. However, some questions also should be explained.

1. Summary section

 ‘the aim of this review’ should not be at the end of the Summary section, and a conclusion should be at the end.

‘GvHD’ should be explained.

Line 22, ‘TGF- , but’ ?

Line 26, delete ‘(Breg)’.

2. Introduction section

Line 44, delete ‘(EAE)’ and ‘(CIA)’.

‘the aim of this review’ should be added in the end paragraph of Introduction section.

3. Diversity of the Breg Universe section

Line 57, ‘TLR’ should be explained.

Line 65, ‘TGF- as’ ?

‘Tab. 1 & Fig. 2’ or ‘Figure 1 and Figure 2’ ?

Figure 1 the color of arrows for upregulation or downregulation may use red or green, but not black.

Line 105, ‘TGF- as’ ?

Figure 1, the relationship between Tryptophan and Kynurenine? Figure 1 should be revised.

GrB and cytokines IL-10, IL-35 or TGF-β are upregulated or downregulated?

4. Regulatory B Cells in Infections section

Line 206, ‘we showed that the HIV protein Nef’, the reference ?

Line 237, delete ‘, TCR = T cell receptor’

5. Conclusions section

Line 278, delete ‘(ATMPs)’

6. There are some related references that are not cited in this review paper.

Li S, Mirlekar B, Johnson BM, Brickey WJ, Wrobel JA, Yang N, Song D, Entwistle S, Tan X, Deng M, Cui Y, Li W, Vincent BG, Gale M Jr, Pylayeva-Gupta Y, Ting JP. STING-induced regulatory B cells compromise NK function in cancer immunity. Nature. 2022;610(7931):373-380.

Zou J, Zeng Z, Xie W, Zeng Z. Immunotherapy with regulatory T and B cells in periodontitis. Int Immunopharmacol. 2022;109:108797.

Li C, Liu P, Yao H, Zhu H, Zhang S, Meng F, Li S, Li G, Peng Y, Gu J, Zhu L, Jiang Y, Dai A. Regulatory B cells protect against chronic hypoxia-induced pulmonary hypertension by modulating the Tfh/Tfr immune balance. Immunology. 2023;168(4):580-596.

Hassuna NA, Hussien SS, Abdelhakeem M, Aboalela A, Ahmed E, Abdelrahim SS. Regulatory B cells (Bregs) in Helicobacter pylori chronic infection. Helicobacter. 2023;28(2):e12951.

Bakhtiar S, Kaffenberger C, Salzmann-Manrique E, Donhauser S, Lueck L, Karaca NE, Gonzalez-Granado LI, Hazar E, Keles S, Seidel MG, Fekadu J, Königs C, Schubert R, Bader P, Huenecke S. Regulatory B cells in patients suffering from inborn errors of immunity with severe immune dysregulation. J Autoimmun. 2022;132:102891.

Kalkal M, Chauhan R, Thakur RS, Tiwari M, Pande V, Das J. IL-10 Producing Regulatory B Cells Mediated Protection against Murine Malaria Pathogenesis. Biology. 2022;11(5):669.

Loisel S, Lansiaux P, Rossille D, Ménard C, Dulong J, Monvoisin C, Bescher N, Bézier I, Latour M, Cras A, Farge D, Tarte K. Regulatory B Cells Contribute to the Clinical Response After Bone Marrow-Derived Mesenchymal Stromal Cell Infusion in Patients With Systemic Sclerosis. Stem Cells Transl Med. 2023;12(4):194-206.

Comments on the Quality of English Language

Extensive editing of English language required.

Author Response

Reviewer #3:

In the present work, Veh et al. try to review regulatory B cells - immunopathological and

prognostic potential in humans. However, some questions also should be explained.

  1. Summary section

‘the aim of this review’ should not be at the end of the Summary section, and a conclusion

should be at the end.

Summary has been modified accordingly.

‘GvHD’ should be explained.

GvHD is now spelled out.

Line 22, ‘TGF- , but’ ?

Corrected to TGF-b

Line 26, delete ‘(Breg)’.

Deleted.

  1. Introduction section

Line 44, delete ‘(EAE)’ and ‘(CIA)’.

Deleted.

‘the aim of this review’ should be added in the end paragraph of Introduction section.

The Introduction Section has been modified accordingly.

  1. Diversity of the Breg Universe section

Line 57, ‘TLR’ should be explained.

TLR has been spelled out.

Line 65, ‘TGF- as’ ?

Corrected to TGF-b

‘Tab. 1 & Fig. 2’ or ‘Figure 1 and Figure 2’ ?

Fig. 2 & Tab. 1

Figure 1 the color of arrows for upregulation or downregulation may use red or green, but not

black.

Color has been added to the arrows

Line 105, ‘TGF- as’ ?

Corrected to TGF-b

Figure 1, the relationship between Tryptophan and Kynurenine? Figure 1 should be revised.

Figure 1 has been revised.

GrB and cytokines IL-10, IL-35 or TGF-β are upregulated or downregulated?

Figure has been revised and arrows have been added.

  1. Regulatory B Cells in Infections section

Line 206, ‘we showed that the HIV protein Nef’, the reference ?

Reference has been added.

Line 237, delete ‘, TCR = T cell receptor’

Deleted

  1. Conclusions section

Line 278, delete ‘(ATMPs)’

Deleted

  1. There are some related references that are not cited in this review paper.

Li S, Mirlekar B, Johnson BM, Brickey WJ, Wrobel JA, Yang N, Song D, Entwistle S, Tan X, Deng

M, Cui Y, Li W, Vincent BG, Gale M Jr, Pylayeva-Gupta Y, Ting JP. STING-induced regulatory B

cells compromise NK function in cancer immunity. Nature. 2022;610(7931):373-380.

Zou J, Zeng Z, Xie W, Zeng Z. Immunotherapy with regulatory T and B cells in periodontitis. Int

Immunopharmacol. 2022;109:108797.

Li C, Liu P, Yao H, Zhu H, Zhang S, Meng F, Li S, Li G, Peng Y, Gu J, Zhu L, Jiang Y, Dai A.

Regulatory B cells protect against chronic hypoxia-induced pulmonary hypertension by

modulating the Tfh/Tfr immune balance. Immunology. 2023;168(4):580-596.

Hassuna NA, Hussien SS, Abdelhakeem M, Aboalela A, Ahmed E, Abdelrahim SS. Regulatory B

cells (Bregs) in Helicobacter pylori chronic infection. Helicobacter. 2023;28(2):e12951.

Bakhtiar S, Kaffenberger C, Salzmann-Manrique E, Donhauser S, Lueck L, Karaca NE,

Gonzalez-Granado LI, Hazar E, Keles S, Seidel MG, Fekadu J, Königs C, Schubert R, Bader P,

Huenecke S. Regulatory B cells in patients suffering from inborn errors of immunity with severe

immune dysregulation. J Autoimmun. 2022;132:102891.

Kalkal M, Chauhan R, Thakur RS, Tiwari M, Pande V, Das J. IL-10 Producing Regulatory B Cells

Mediated Protection against Murine Malaria Pathogenesis. Biology. 2022;11(5):669.

Loisel S, Lansiaux P, Rossille D, Ménard C, Dulong J, Monvoisin C, Bescher N, Bézier I, Latour

M, Cras A, Farge D, Tarte K. Regulatory B Cells Contribute to the Clinical Response After Bone

Marrow-Derived Mesenchymal Stromal Cell Infusion in Patients With Systemic Sclerosis. Stem

Cells Transl Med. 2023;12(4):194-206.

All references with direct relevance for the human immune system have been included.

Reviewer 4 Report

Comments and Suggestions for Authors

Johanna et al. narratively reviewed the Bregs in several human diseases. 

Please refer to my comments: 

1) Please specify your affiliation (refer to the journal's template)

2) Typing error or formating error: TGF-β, the β is missing. It also affected the legend (line 92).

3) Avoid abbreviations in the Abstract. Please define them

4) Why not go for a scoping or systematic review? How to avoid reporting bias in this review? 

5) Fig 1: Kindly explain the meaning of "arrow down". Kindly convert "Anergie" into English. 

6) Fig 2: The arrow over Kynurenine is missing. Please specify if the cytokine that is released or affecting the cell activation.  

7) Table 1 and 2: Please prepare the tables according to the journal requirement. Make sure the references are matched with the reference list. Highly recommended to separate Table 2 into shorter tables for each disease area (Inflammation, Infections, cancer). 

8) Please support your review with completed or ongoing clinical trials (for example https://clinicaltrials.gov/). Please state it out if no completed or ongoing trial so far. This is crucial to show the research gap.

9) Briefly discuss more on the TGF-β subtypes on Breg. 

10) Briefly discuss more on the Breg, IL10 and IgG4 cross-talk in immunosuppression or pathogenesis of human diseases. 

11) Do not forget to complete the "author contributions", "Funding" and "Conflicts of interest" sessions. 

12) Correct the reference list format. 

13) There are plenty of similar reviews on this topic. What are the strengths and limitations of your review? What are the additional impacts or benefits from your review, as compared to others? 

Author Response

Reviewer #4:

Johanna et al. narratively reviewed the Bregs in several human diseases.

Please refer to my comments:

1) Please specify your affiliation (refer to the journal's template)

Affiliation is correct.

2) Typing error or formating error: TGF-β, the β is missing. It also affected the legend (line 92).

Corrected to TGF-b

3) Avoid abbreviations in the Abstract. Please define them

Abbreviations have in spelled out in the Abstract

4) Why not go for a scoping or systematic review? How to avoid reporting bias in this review?

The present review is invited and is a translation of an already existing review. Major changes have not been granted by the publisher of the original article. Apart from that, a reporting bias can never be completely ruled out in a review. However, we tried to write it as objectively and comprehensively as possible.

5) Fig 1: Kindly explain the meaning of "arrow down". Kindly convert "Anergie" into English.

Corrected to Anergy

6) Fig 2: The arrow over Kynurenine is missing. Please specify if the cytokine that is released or

affecting the cell activation.

Figure has been revised accordingly and arrow have been added

7) Table 1 and 2: Please prepare the tables according to the journal requirement. Make sure the

references are matched with the reference list. Highly recommended to separate Table 2 into

shorter tables for each disease area (Inflammation, Infections, cancer).

Letter size in tables has been increased, table 2 has been split in 5 sub-tables. If letter size is still too small tables could be turned into an upright position. References are matched.

8) Please support your review with completed or ongoing clinical trials (for example

https://clinicaltrials.gov/). Please state it out if no completed or ongoing trial so far. This is crucial

to show the research gap.

Information on clinical trials has been added to the Conclusion section.

9) Briefly discuss more on the TGF-β subtypes on Breg.

Brief paragraph on TGF-b Breg subtypes has been added to the Universe Section.

10) Briefly discuss more on the Breg, IL10 and IgG4 cross-talk in immunosuppression or

pathogenesis of human diseases.

Brief paragraph on Breg, IL-10 and IgG4 cross-talk has been added to the Universe Section.

11) Do not forget to complete the "author contributions", "Funding" and "Conflicts of interest"

sessions.

"author contributions", "Funding" and "Conflicts of interest" Sections have been completed.

12) Correct the reference list format.

Reference list format corrected.

13) There are plenty of similar reviews on this topic. What are the strengths and limitations of

your review? What are the additional impacts or benefits from your review, as compared to

others?

The present review is invited and is a translation of an already existing review. We tried to write it as concisely, objectively and comprehensively as possible. In addition, our review focuses on human relevance instead of overloading it with too much information from various animal model, which may or may not be relevant for the human immune system.

Round 2

Reviewer 3 Report

Comments and Suggestions for Authors

Thanks for author’s responses. However, the format of tables may be revised according to Journal style.

Comments on the Quality of English Language

Minor editing of English language required.

Author Response

Reviewer #3:

Thanks for author’s responses. However, the format of tables may be revised according to Journal style.

In addition to the manuscript pdf-file for reviewing purposes, all tables have been provided in MS-Excel format and can be modified and inserted as needed by the production office. If the production office specifies required changes to the tables in more detail to us we will be happy to assist.

Minor editing of English language required.

Three of the four reviewers assigned to the manuscript considered the English language of our review “fine with no issues detected”. Furthermore, I have been publishing peer-reviewed manuscripts in english language for more than 20 years, in part during my 4-year postdoctoral fellowship in the United States, without ever having been pointed to insufficient English in my manuscripts. Lastly, the current review was proof-read by a native and fluent US-English speaker. We therefore do not see the necessity to use an additional professional editing service.

Reviewer 4 Report

Comments and Suggestions for Authors

Thank you the authors for the amendment. 

1) Please emphasize the importance and impact of this review in the Introduction in comparison to previous similar reviews. 

2) Please be specific how the authors avoid reporting bias in preparing this review. 

Thanks

Author Response

1) Please emphasize the importance and impact of this review in the Introduction in comparison to previous similar reviews. 

As explained further below, the current review focuses on human relevance instead of commingling findings from humans with findings from animal models, which may or may not be relevant for the human immune system. We added this information into the Introduction Section of our manuscript.

2) Please be specific how the authors avoid reporting bias in preparing this review. 

Before starting to prepare the manuscript, we read and analyzed a series (>10) of current reviews in the field. After having identifying the reporting gaps of these reviews, we tried to compensate them by focusing our own review on identified gaps to the best of our knowledge. One of the most frequently identified gaps in other reviews was the commingling of findings in humans and findings in animal models. We therefore focused our own review on the human immune system and corresponding findings in humans. Nevertheless, we are aware of the fact, that reporting biases can never be ruled out in a review and we were most likely not able to completely avoid reporting bias in our review.